# Melorheostosis: A Review of the Literature and a Case Report

**DOI:** 10.3390/medicina59050869

**Published:** 2023-04-30

**Authors:** Sergiu Iordache, Adrian Cursaru, Bogdan Serban, Mihai Costache, Razvan Spiridonica, Bogdan Cretu, Catalin Cirstoiu

**Affiliations:** Department Orthopedic & Traumatology, Carol Davila University Medicine & Pharmacy, University Emergency Hospital, 050098 Bucharest, Romania

**Keywords:** melorheostosis, Leri’s disease, resection reconstruction, bone tumors

## Abstract

*Background and Objectives*: Melorheostosis, also referred to in the literature as Leri’s disease, is an unusual mesenchymal dysplasia with the clinical appearance of benign sclerosing bone dysplasia; it frequently occurs in late adolescence. Any bone in the skeletal system can be affected by this disease, though the long bones of the lower extremities are the most common, at any age. Melorheostosis has a chronic evolution, and symptoms are usually absent in the early stages. The etiopathogenesis is still unknown, however, numerous theories have been proposed that could explain the appearance of this lesion formation. An association with other benign or malignant bone lesions is also possible, and associations with osteosarcoma, malignant fibrous histiocytoma, or Buschke–Ollendorff syndrome have also been reported. There have also been reported cases of the malignant transformation of a pre-existing melorheostosis lesion into malignant fibrous histiocytoma or osteosarcoma. The diagnosis of melorheostosis can be made only based on radiological images, but, due to its polymorphism, additional imaging investigations are often necessary and sometimes only a biopsy can establish a definite diagnosis. Because there are currently no guidelines for treatment based on scientific evidence, due to the low number of cases diagnosed worldwide, our objective was to highlight the early recognition and specific surgical treatments for better prognosis and outcomes. *Materials and Methods*: We conducted a review of the literature consisting of original papers, case reports, and case series and presented the clinical and paraclinical characteristics of melorheostosis. We aimed to synthesize the treatment methods available in the literature as well as determine possible future directions related to the treatment of melorheostosis. Furthermore, we presented the results of a case of femoral melorheostosis admitted to the orthopedics department of the University Emergency Hospital of Bucharest in a 46-year-old female patient with severe pain in the left thigh and limitation of joint mobility. Following the clinical examination, the patient complained of pain in the middle third of the left thigh in the antero-medial compartment; the pain appeared spontaneously and was aggravated during physical activity. The pain started about two years prior, but the patient experienced complete pain relief after the administration of non-steroidal anti-inflammatory drugs. In the last six months, the patient presented an increase in pain intensity without significant improvement following the administration of non-steroidal anti-inflammatory drugs. The patient’s symptoms were mainly determined by the increase in the volume of the tumor and the mass effect on the adjacent tissues, especially on the vessels and the femoral nerve. The CT examination and bone scintigraphy showed a unique lesion in the middle third of the left femur and no oncological changes in the thoracic, abdominal, and pelvic regions; however, at the level of the femoral shaft, there was a localized cortical and pericortical bone lesion formation that surrounded approximately 180 degrees of the femoral shaft (anterior, medial, and lateral). It had a predominantly sclerotic structure but was associated with lytic areas with thickening of the bone cortex and areas of periosteal reaction. The next therapeutic gesture was to perform an incisional biopsy using a lateral approach at the level of the thigh. The histopathological result supported the diagnosis of melorheostosis. Additionally, immunohistochemical tests completed the data obtained after the microscopic examination through the classic histopathological technique The patient was discharged and included in a full medical recovery program for eight weeks in a specialized medical center, during which she also received analgesic treatment in maximum doses, but without improvement regarding her symptoms. Taking into account the chronic evolution of the pain, the complete lack of response to conservative treatment after eight weeks, and the lack of treatment guidelines in the case of melorheostosis, a surgical approach needed to be considered. The surgical option in this case, considering the circumferential location of the lesion at the level of the femoral diaphysis, was a radical resection. The surgical approach consisted of segmental resection to healthy bone tissue and reconstruction of the remaining defect with a modular tumoral prosthesis. At the 45-day postoperative control, the patient no longer complained of pain in the operated-on limb and was mobile with full support without gait difficulties. The follow-up period was one year, and the patient presented complete pain relief and a very good functional outcome. *Results:* In the case of asymptomatic patients, conservative treatment seems to be a good option with optimal results. However, for benign tumors, it remains unclear whether radical surgery is a viable option. *Conclusions*: Melorheostosis remains an incompletely understood disease, given the limited number of cases worldwide, and thus, there is a lack of clinical guidelines regarding specialized treatment.

## 1. Introduction

Melorheostosis, which is also known in the literature as Leri’s disease, is an unusual mesenchymal dysplasia with the clinical appearance of benign sclerosing bone dysplasia; it frequently occurs in late adolescence. “Melorheostosis” is a term that is derived from the Greek “melos”, meaning limb, “rhein”, meaning flowing, and “ostosis”, meaning bone formation. Leri and Joanny described this disease for the first time in 1922 and it is still not well understood. The density distribution of melorheostosis, like the segmental distribution, can correspond with the anatomical distribution of nerve roots or blood vessels [1].

Any bone in the skeletal system can be affected by this disease, but the long bones of the lower extremities are the most common, and patients can be affected at any age [1]. The osseous involvement can be polyostotic or monostotic, but most of the time it is confined to one limb; occasionally it may be bilateral. Early presentation and multiple limb involvement may predict a poorer prognosis in terms of complications.

However, the diagnosis is often established in early or late adulthood due to the disease’s silent clinical manifestations; the disease remains asymptomatic for a long period, often being discovered only incidentally [1], or present with pain, swelling, deformities, contractures, muscle atrophy, and joint stiffness. Soft tissue fibrosis with ligament and tendon retraction may be observed, and it is not uncommon to see equinovarus, valgus, or varus foot deformities. Due to the asymmetric early fusion of epiphyses, we can often see a limb length discrepancy.

Melorheostosis may mimic other conditions, such as osteoma, myositis ossificans, or parosteal osteosarcoma [2]. The prevalence is equal for both sexes, and approximately 50% of diagnoses are established around the age of 20 years [2]. Despite the fact that it is a disease described about 100 years ago, up to now only about 400 cases of melorostosis have been reported.

The etiopathogenesis is still unknown, with numerous theories being proposed that could explain the appearance of this lesion. Recent molecular biology studies have shown mutations in the MAP2K1 (Mitogen-Activated Protein Kinase Kinase 1) gene in dripping candle wax forms and mutations in SMAD3 in endosteal forms [3].

In the literature, it is reported that isolated melorheostosis is associated with random somatic mutations of the MAP2K1 gene. This gene encodes the MEK1 protein kinase, which is part of the RAS/MAPK signaling cascade. Mutations in this pathway often lead to malignancy. In addition, isolated somatic mutations of MAP2K1 can lead to the localized benign proliferation of bone cells which results in melorheostosis. However, MAP2K1 mutation can disrupt the bone morphogenetic protein-2 (BMP2) osteoblast-mediated mineralization process which also leads to significant unmineralized osteoid formation [4].

The MAP2K1 oncogene is significant in the bone formation of humans and opens the potential treatment of melorheostosis by gene therapy in the future [1].

A review of 23 cases conducted by Freyschmidt stated that the cause of the disease is unknown, and the concept of mosaicism was proposed as an explanation for the disease’s sporadic occurrence, its asymmetric “segmental” pattern with the variable extent of involvement, and its equal gender ratio [5].

LEM domain-containing protein 3 (LEMD3) gene mutations have been described in several familial cases of melorheostosis; there was no direct correlation with this pathology, but they were directly associated with other hereditary dysplasias such as osteopoikilosis (OPK), a melorheostosis-associated disease presenting with a hyperostosis phenomenon similar to melorheostosis [4].

As melorheostosis often coexists with OPK or within a family with LEMD3 mutations, the TGF-β/SMAD pathway may also contribute to melorheostosis pathogenesis. The TGF-β/SMAD pathway is crucial for skeletal embryonic development and postnatal homeostasis. Dysregulation of the TGF-β signaling pathway is associated with a spectrum of osseous defects as seen in several dominant genetic disorders: Marfan syndrome and Loeys–Dietz syndrome [6].

Melorheostosis can be confirmed by radiological studies such as X-rays, CTs, MRIs, and bone scans. Bone scintigraphy, MRI, or CT scans could help the physician to decide which surgical treatment is better [7].

In this review and case presentation, we propose to summarize what is known regarding the clinico-radiological features, pathophysiology, and management of this rare bone disease, and to present what we have found.

## 2. Case Presentation

We present the case of a 46-year-old female patient who presented to the orthopedics and traumatology department of the University Emergency Hospital of Bucharest with severe pain in the left thigh and limitation of joint mobility. Following the clinical examination, the patient complained of pain in the middle third of the left thigh in the antero-medial part; the pain appeared spontaneously and was aggravated during physical activity. The pain started about two years ago, but the patient experienced complete pain relief after the administration of non-steroidal anti-inflammatory drugs. In the last six months, the patient presented an increase in pain intensity without significant improvement following the administration of non-steroidal anti-inflammatory drugs. The patient used non-steroidal anti-inflammatory drugs daily, without a significant improvement in pain, which significantly affected her quality of life. No shortening of the limbs or stiffness in the adjacent joints was observed. At the local examination, the patient had normal skin, without local changes, without inflammatory signs present, and no signs of ischemia. An X-ray of the femur in the antero-posterior and lateral incidence was performed which identified a unique bone lesion with dimensions of approximately 12/10 cm in the coronal plane located on the antero-medial cortex of the femur with a mixed appearance (Figure 1 and Figure 2). After that, we performed a whole-body scintigraphy which highlighted the same bone lesion described in the radiological examination with moderate uptake in the late phase, without highlighting other associated bone lesions.

Furthermore, we conducted a review of the literature consisting of original papers, case reports, and case series and presented the clinical and paraclinical characteristics of melorheostosis. We synthesized not only the treatment methods available in the literature but also possible future directions related to the treatment of melorheostosis.

At the time of admission to the orthopedic department, biologically speaking, the patient had no pathological values in the blood count, alkaline phosphatase, or inflammatory markers (Table 1).

Considering the size of the lesion, the imaging aspect, and the clinical symptomatology, a differential diagnosis with parosteal osteosarcoma was required. A CT scan of the thorax, abdomen, pelvis, and left thigh was performed. The CT examination showed no oncological changes in the thoracic, abdominal, and pelvic regions; however, on the femoral shaft, there was a localized cortical and pericortical bone lesion formation that surrounded approximately 180 degrees of the femoral shaft (anterior, medial, and lateral). It had a predominantly sclerotic structure but was associated with lytic areas with thickening of the bone cortex and areas of periosteal reaction (Figure 3).

Whole-body-scan bone scintigraphy was performed and showed the same bone lesion formation that demonstrated moderate uptake of the radiotracer only in the late phase (Figure 4).

Considering the size of the tumor, location, and radiological appearance, there was a high risk for a malignant lesion. The management of a malignant pathology requires rigorous preoperative planning that includes the choice of the surgical approach for the incisional biopsy so as to minimize the spread of tumor cells and establish the level of resection while taking into account the oncological margins, the possibilities of reconstruction, and the viable alternatives to cover the implant in case of tumor invasion in the adjacent soft tissues that will require surgical excision. The final therapeutic decision took into account all of the aspects listed above and the patient was presented with all the potential risks and complications that may arise. The next therapeutic gesture was the performance of an incisional biopsy using a lateral approach at the level of the thigh to allow, depending on the histopathological result, the performance of a second curative surgical intervention.

The histopathological result supported the diagnosis of melorheostosis (Figure 5, Figure 6, Figure 7 and Figure 8). Additionally, immunohistochemical tests completed the data obtained after the microscopic examination through the classic histopathological technique. CD45/LCA (clone PD7/26/16 and 2B11, Biocare) demonstrated the presence of inflammatory cells, but overall, the rest of the immunohistochemical tests were non-specific and did not reveal the presence of epithelial or other tumor proliferations on the analyzed specimens. CD138 (clone B-A38, Biocare) showed rare plasma cells dispersed in hematoforming marrow, CD56 (clone BC56C04, Biocare) highlighted few osteoblasts, and all specific immunomarkers for cytokeratins were negative (Pan Cytokeratin AE1/AE3 clone AE1/AE3 and CK8/18 clone CK8/18 both from Biocare). Ki67 (clone SP6, Biocare) was negative in areas of reactive fibrosis and positive in a few cells in the hematoform marrow, suggesting a benign lesion. CD56 usually highlights neuroendocrine tumors, myeloma, myeloid leukemia, and Nk/T cell lymphomas and is also positive in some rare sarcomas. The negative panCK marker denied the possible epithelial tumor origin. Although there are no specific immunomarkers for this pathological entity, there are some studies that try to find correlations between the immunohistochemical expression of some proteins and the severity of the lesions. One study used immunohistochemistry to investigate the expression of several proteins in the affected bone tissues of melorheostosis patients. The study found that there was increased expression of certain proteins, including transforming growth factor beta (TGF-β), bone morphogenetic protein 2 (BMP-2), and insulin-like growth factor 1 (IGF-1) in the bone tissues of melorheostosis patients compared to healthy controls [8].

The increased expression of these proteins suggests that they may play a role in the pathogenesis of melorheostosis by promoting abnormal bone growth. Further studies using immunohistochemistry may help us to better understand the molecular mechanisms underlying melorheostosis and identify potential therapeutic targets for this rare bone disorder.

The patient was discharged with the recommendations of analgesic and anti-inflammatory treatment in high doses, and she was included in a medical recovery program. The patient was included in the full medical recovery program for eight weeks in a specialized medical center, during which she also received anti-analgesic treatment in maximum doses, but without improvement regarding symptoms. Taking into account the chronic evolution of the pain, the complete lack of response to conservative treatment after eight weeks, and the lack of treatment guidelines in the case of melorheostosis, a surgical approach needed to be considered. Thus, the patient returned to the orthopedic department with significant pain, gait deformation, and a decreased ability to flex the thigh and extend the knee. Considering the size of the lesion, its location, with the increased potential to cause compression of the vessels and the femoral nerve, the lack of changes in the soft tissues, and the symptoms not being responsive to conservative treatment, as well as a histopathological result of a benign lesion, choosing the best treatment was a challenge for the surgical team. Although it was a benign lesion, the important impact on the quality of life together with the failure of non-surgical treatment allowed the consideration of surgical treatment. The patient’s symptoms were mainly determined by the increase in the volume of the tumor and the mass effect on the adjacent tissues, especially on the vessels and the femoral nerve. The surgical option in this case, considering the circumferential location of the lesion at the level of the femoral diaphysis, was a radical one. The surgical approach consisted of segmental resection to healthy bone tissue and reconstruction of the remaining defect with a modular tumoral prosthesis. After adequate preoperative preparation, surgery was performed, and, using a lateral iterative approach, the bone lesion was resected to the normal macroscopic bone tissue (Figure 9). The proximal, distal, medial, lateral, anterior, and posterior resection limits were sent for extemporaneous pathological examination; all of them were negative for tumor invasion. The reconstruction was carried out using a modular diaphyseal segment fixed intramedullary using 10 cm femoral stems (Figure 10). In the preoperative planning, the potential surgical difficulties were related to the correct restoration of the length of the limb, the correct establishment of the rotation of the femur, and obtaining a minimum of 10 cm of healthy bone in the proximal and distal femur after lesion resection so as to allow a good fixation of the implant. The surgery was performed without unique issues, and the postoperative results were optimal with no leg length discrepancy. Taking into account the magnitude of the surgical intervention for this type of benign lesion, the risks, such as deep venous thrombosis, pulmonary thromboembolism, intraoperative vascular-nerve injuries as well as considering the intimate contact of the lesion with vessels and nerves, leg length discrepancy, and the major septic risk, should be remembered. At the 45-day postoperative control, the patient no longer complained of pain in the operated-on limb and was mobile with full support without gait difficulties.

It was recommended that the patient continue the recovery program and periodic reassessment through the orthopedic department. The follow-up period was one year, and the patient presented complete pain relief and a very good functional outcome with no gait limitations.

## 3. Discussion

There are currently no guidelines for treatment based on scientific evidence given the low number of cases diagnosed worldwide. Therefore, treatment must be adapted to each patient and requires a multidisciplinary team. Treatment decisions must be made based on the severity of the symptoms and the therapeutic possibilities.

Given the benign nature of melorheostosis and the extent of the lesion in the soft tissues, its size, the presence of symptoms, and the possibility of resection and reconstruction, in many cases, non-surgical treatment is sufficient to improve symptoms and restore the function of the limb.

Clinical manifestations may initially be absent, but, with the progression of the disease, symptoms have been reported that are related to the local evolution of the lesion and the mass effect it produces on soft tissues, such as local pain, pathological muscle contractures, and joint stiffness. In children, an altered bone structure with deformity and shortening of the limbs may be the first sign. Damage to the axial skeleton is exceptional, frequently involving the appendicular skeleton, especially the long bones of the lower limbs [9]. Associations of melorheostosis with vascular malformations and nerve complications due to the mass effect of the lesion size with nerve compression syndrome of the peripheral or central nerves have also been reported [5,10]. In the case of tumors with the significant involvement of adjacent soft tissues, signs of subcutaneous fibrosis, fibroids, local edema, hypertrichosis, and fibrolipomas have been reported [11]. An association with other benign or malignant bone lesions is also possible, and associations with osteosarcoma, malignant fibrous histiocytoma, or Buschke–Ollendorff syndrome have been reported [12,13,14]. There have also been reported cases of the malignant transformation of a pre-existing melorheostosis lesion into malignant fibrous histiocytoma or osteosarcoma [15,16].

The diagnosis of melorheostosis can be made only by radiological images, but, due to its polymorphism, additional imaging investigations are frequently necessary, and sometimes only a biopsy can establish a definitive diagnosis.

Five radiological patterns are described, and the characteristic sign consists of periosteal cortical thickening with thick undulating ridges of bone, reminiscent of molten wax (“the dripping candle wax sign”) [1]. Atypical cases consist of osteopathic striate-like, myositis ossificans-like, osteoma-like, or mixed patterns [2]. The CT images are similar to radiographic images and show cortical hyperplasia with areas of hyperdensity in the cortex with a lack of osteolysis areas.

Nuclear magnetic resonance is not a routine investigation used in the diagnosis of melorheostosis but it may reveal bone marrow invasion, especially in forms with predominantly endosteal development or soft tissue status adjacent to the lesion [17,18].

In a study that analyzed 40 patients, it was found that CT scans are a valuable tool for correctly diagnosing bone and articular involvement, and MRI reveals important soft tissue lesions [19].

Bone scintigraphy is an imaging investigation with an important role in highlighting bone metabolic activity; it has been used in the process of diagnosing patients with suspected melorheostosis [20,21]. The scintigraphic characteristic of melorheostosis is moderate uptake in the late phases [22].

The histopathological appearance varies, but several pathognomonic changes have been identified, such as increased bone cortical density, the presence of woven bone features, hypervascular features, an increased number of Haversian systems, and irregular bone growth into the medullary cavity; moreover, newly deposited unmineralized osteoids were seen in affected lesions [23].

Differential diagnosis is required depending on the radiological aspect with osteoma, myositis ossificans, parosteal osteosarcoma, Caffey’s disease, mixed metastases specific to prostate or breast tumors, hypertrophic osteoarthropathy, and focal scleroderma.

Melorheostosis has a chronic evolution, and the symptoms are usually absent in the early stages. Complications are related to lesion extension into adjacent soft tissues that can cause compression phenomena and complications related to bone deformity and damage to the bone structure. The risk of secondary malignant transformation has also been reported in other case reports [24].

Nonsteroidal anti-inflammatory drugs and physical therapy are the first step in the treatment of melorheostosis and have inconsistent results depending on the type and extent of the bone lesion. Other alternative therapies, such as nerve blocking, serial casting, manipulations, and sympathectomies, have also been used [25].

Slimani et al. and Hollick R. J. et al. reported the successful treatment of the pain syndrome in melorheostosis using zolendronate, which also led to an improvement in the evolution following scintigraphic monitoring [12,26].

A significant clinical improvement in symptomatology was presented by Byberg S. et al. in a case report of a patient with monostatic melorheostosis following the administration of denosumab [27].

Surgical treatment can be divided into adjuvant surgical treatment aimed at ameliorating the complications of the disease and treatment with a curative function, which consists of the resectioning of the bone lesion and reconstruction.

The primary role of adjuvant surgical treatment is in improving the patient’s quality of life and restoring the function of the affected bone segment. Among the adjuvant procedures, it is worth mentioning tendon lengthening interventions, the excision of fibrous tissue, osteotomies, capsulotomies, fasciotomies, arthrodesis, and tendon elongations [28,29]. Younge D. et al. recommend performing wide capsulotomies and tenotomies after correcting limb deformities to the detriment of simple tendon elongations [30].

In a case report, John B. et al. presented the results of a patient with melorheostosis who benefited from repeated interventions for the excision of fibrous tissue in the knee, which had favorable results on the functionality of the affected limb [31].

External fixators have also been used successfully in the treatment of limb inequalities and muscle contractures specific to melorheostosis [32,33].

However, adjuvant surgical treatments address the consequences of this pathology and do not have a curative purpose, so, in many cases, multiple surgeries with unsatisfactory long-term results are required.

In some particular cases, due to massive extension into the adjacent tissues and the complications related to the mass effect on the vascular and nervous structures, radical interventions such as the resection and reconstruction of the bone segment with tumor prostheses, arthroplasty, or even amputation may be required [34,35,36]. Possible complications of the surgery in our case are related to the disposition of the lesion in intimate contact with the vessels and the femoral nerve, the possibility of restoring limb length, and prosthetic difficulties, considering that at least 10 cm of intact bone was needed for fixation in the proximal segment.

Cases of melorheostosis without the major involvement of adjacent soft tissues and those that are located at the level of long bones can be treated surgically with resection and reconstruction using primary prostheses, tumor prostheses, or various osoteosynthesis materials.

## 4. Conclusions

Given the limited number of cases worldwide and thus, the lack of clinical guidelines regarding specialized treatment, melorheostosis remains an incompletely understood disease. In the case of asymptomatic patients, conservative treatment seems to be a good option with optimal results. Patients with severe symptoms and significantly limited joint mobility should benefit from surgery for both curative and quality-of-life purposes. However, for benign bone lesions, it remains unclear whether radical surgery is a viable option.

On the other hand, the delay of a possible surgical resection of a tumor can lead to complications that are difficult to treat surgically; the extension of the tumor into the surrounding soft tissues means that the only surgical solution is a radical one, such as amputation or disarticulation.

Another important aspect of the therapeutic decision is represented by the association of melorostosis with other benign or malignant bone tumors. The association with malignant bone tumors such as osteosarcoma or malignant fibrous histiocytoma exponentially increases the mortality or the requirement of radical surgical interventions such as amputation or disarticulation.

The peculiarity of the case consists of the anatomical disposition of the bone lesion with the risk of compression of the vessels and the femoral nerve as well as the lack of complications of the soft tissues, the lack of response to conservative treatment, and the possibility of performing curative surgery without major implications.

## Figures and Tables

**Figure 1 medicina-59-00869-f001:**
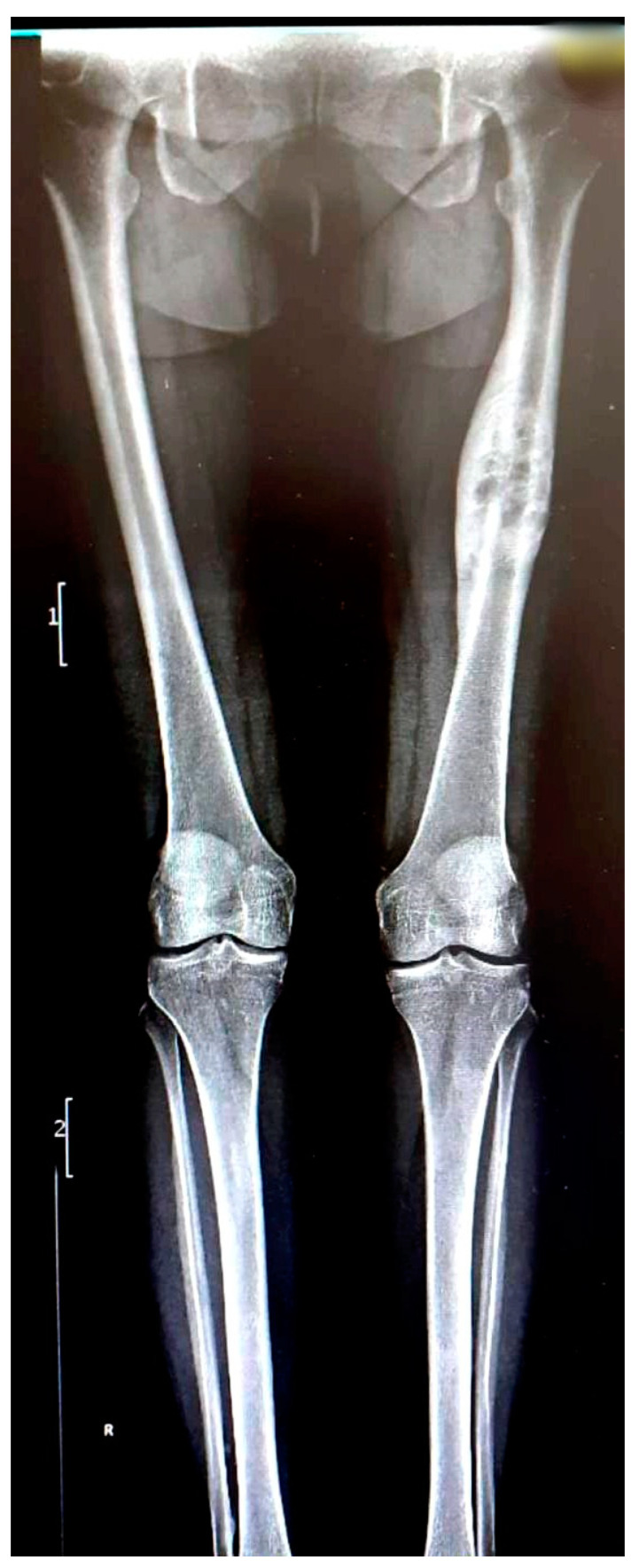
Full-leg X-ray in the antero-posterior view showing a sclerotic bone lesion (12 cm) associated with lytic areas with thickening of the bone cortex.

**Figure 2 medicina-59-00869-f002:**
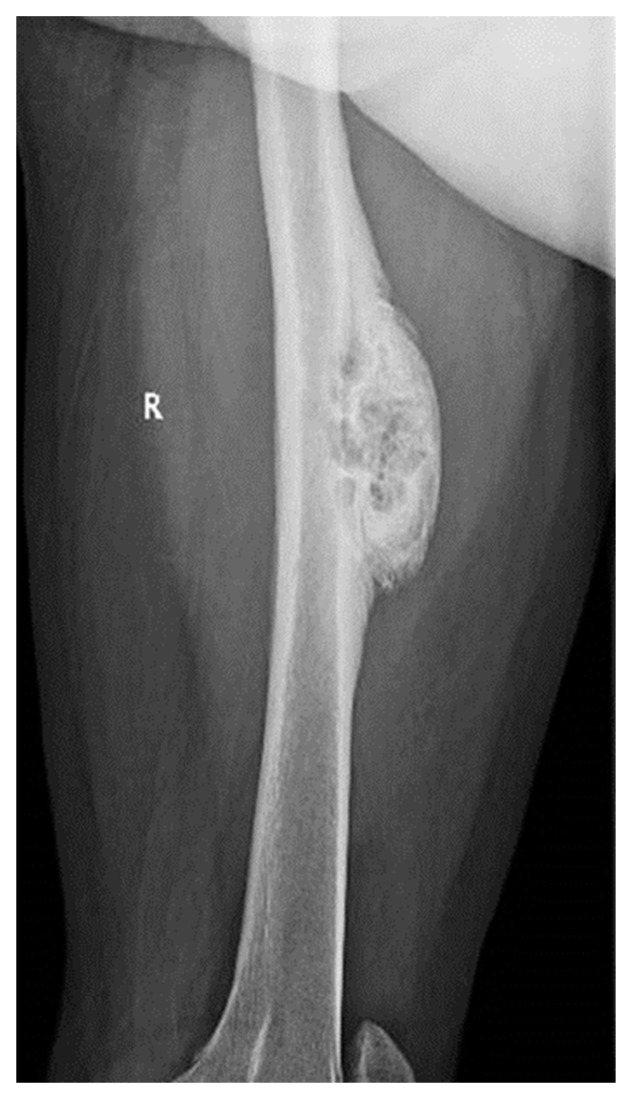
Lateral view X-ray of the femur showing sclerotic bone associated with lytic areas with thickening of the bone cortex.

**Figure 3 medicina-59-00869-f003:**
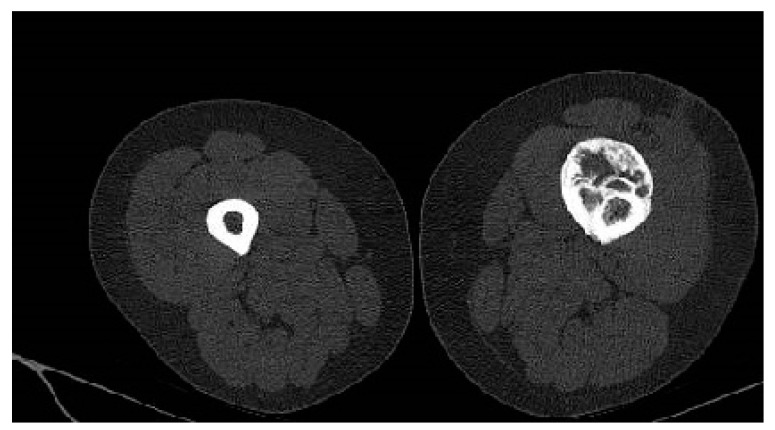
Axial CT scan showing a sclerotic bone with lytic areas of thickening of the bone cortex and areas of periosteal reaction.

**Figure 4 medicina-59-00869-f004:**
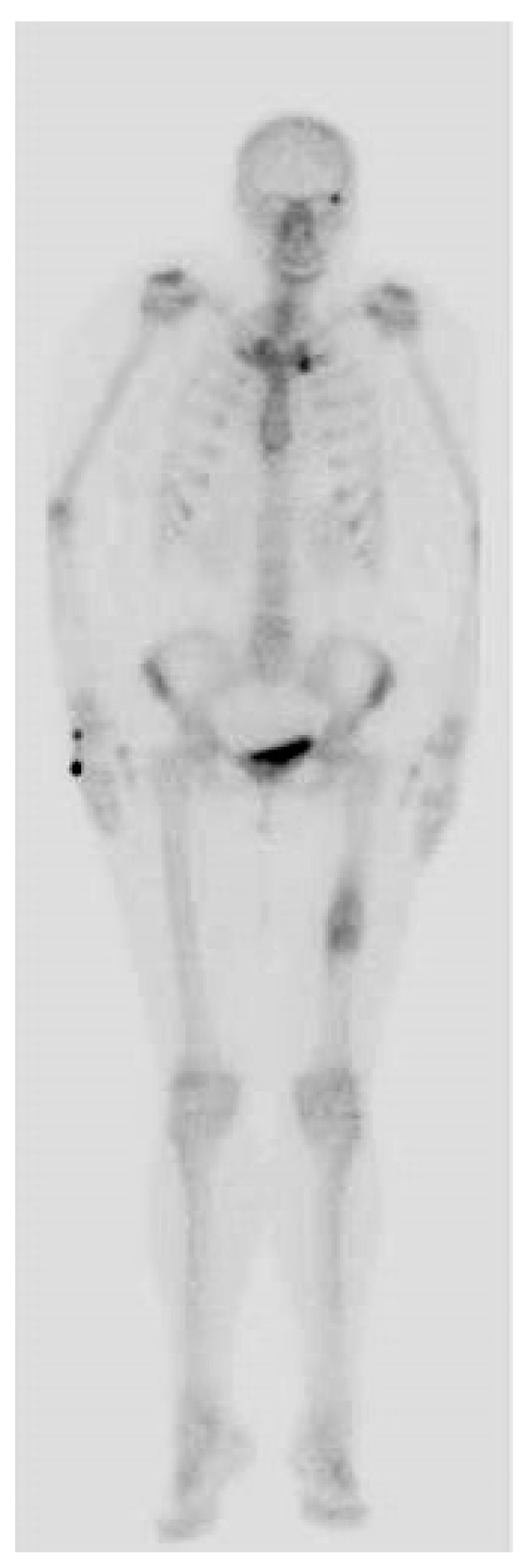
Bone scintigraphy highlighting the bone lesion in the middle third of the left femur with moderate uptake in the late phase.

**Figure 5 medicina-59-00869-f005:**
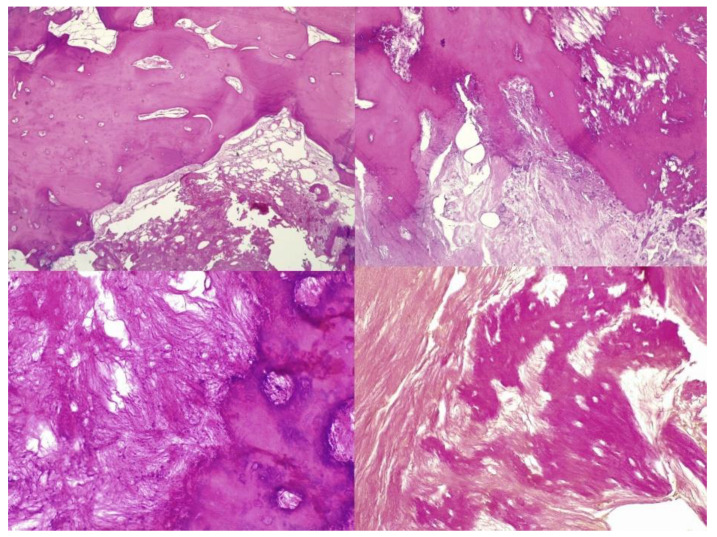
Marked endosteal sclerosis with marrow fibrosis.

**Figure 6 medicina-59-00869-f006:**
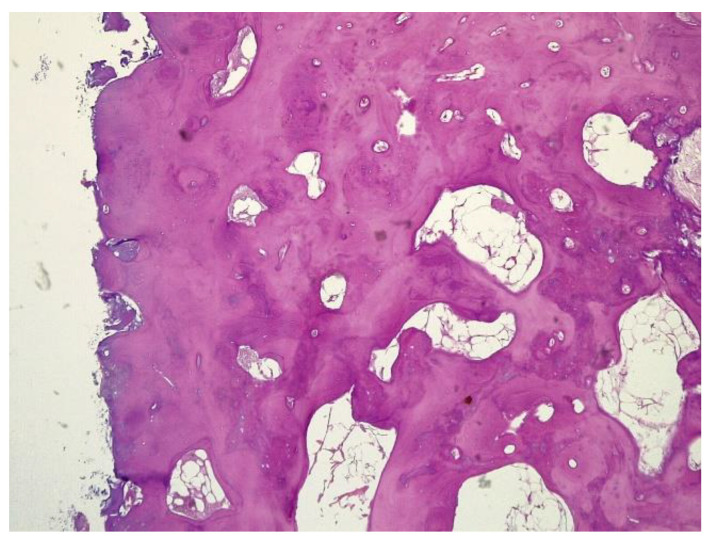
Abnormal proliferation of thickened compact, haversian, or woven bone distorting the normal smooth contour of the periosteal surface of the bone.

**Figure 7 medicina-59-00869-f007:**
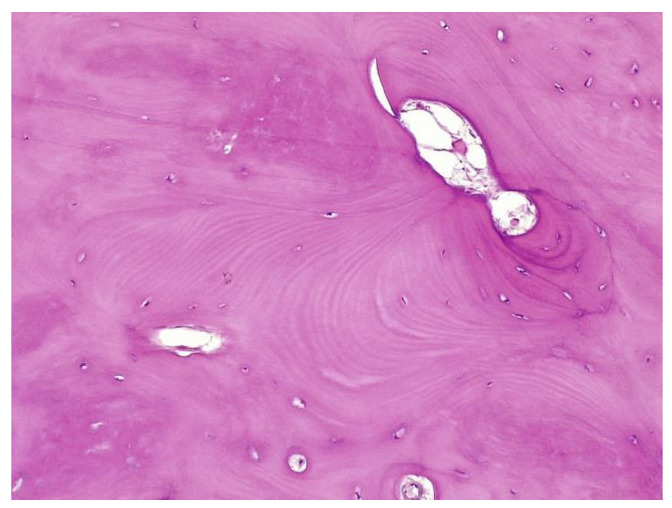
Remarkably dense compact bone with no significant architectural alteration, consistent with melorheostosis.

**Figure 8 medicina-59-00869-f008:**
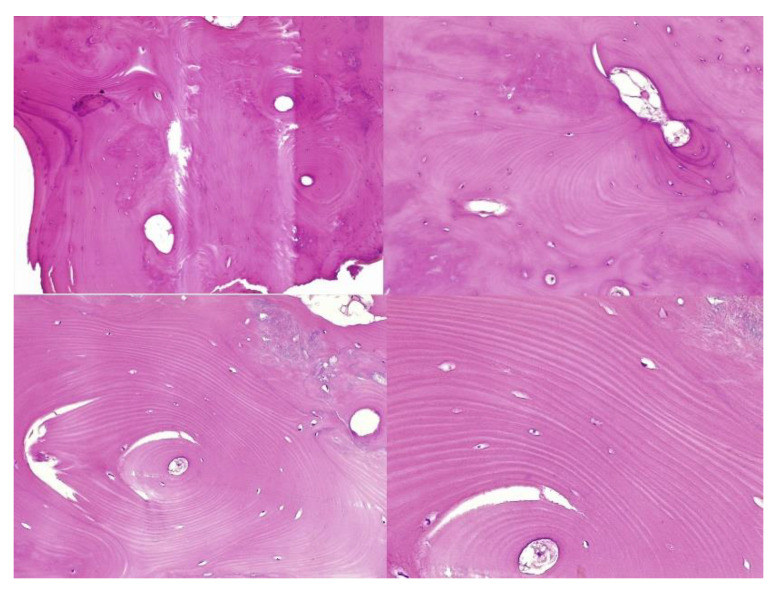
The osseous nodule consisted of mildly hypercellular compact lamellar bone with slightly irregular cement lines.

**Figure 9 medicina-59-00869-f009:**
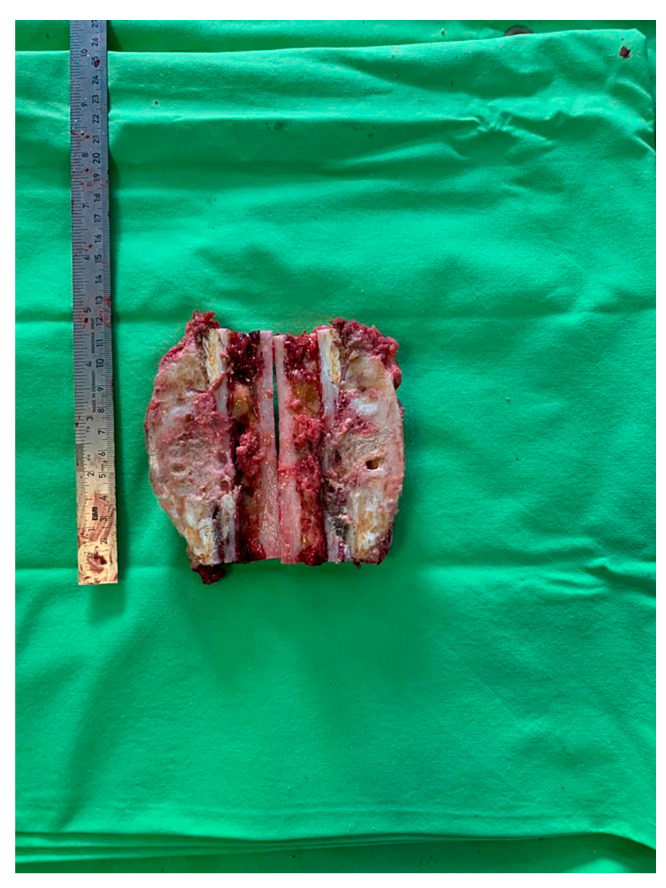
Intraoperative macroscopic image of the resected bone lesion.

**Figure 10 medicina-59-00869-f010:**
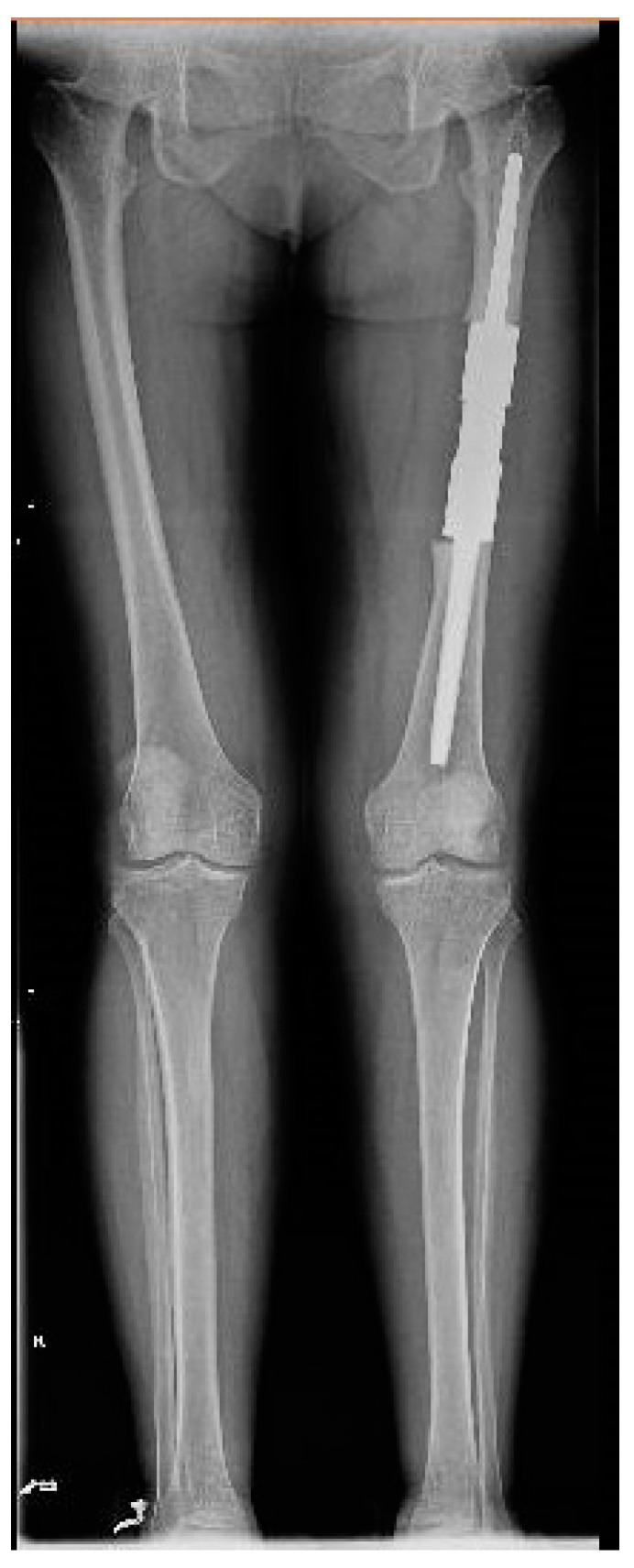
Full-leg X-ray 45 days after surgery.

**Table 1 medicina-59-00869-t001:** Main lab values of the patient at the moment of admission to the hospital.

Biochemical Data	Patient’s Value	Normal Range
WBC (white blood cells)	7.0	3.8–11.8/10^3^/μL
HGB (hemoglobin)	12.4	10.9–14.3 g/dL
PLT (platelets)	251	179–408/10^3^/μL
Fibrinogen	374	238–498 mg/dL
ALKP (alkaline phosphatase)	64	40–136 U/L
FE (iron)	79	50–160 mg/dL
CRP (C-reactive protein)	4.66	0–5 mg/L
ESR (erythrocyte sedimentation rate)	6	5–10 mm/h
Albumin	4.3	3.4–5.2 g/dL
Serum calcium	9.22	8.2–10.7 mg/dL

## Data Availability

Further data concerning the study can be obtained by contacting the corresponding author.

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
