# Peer review of "Melorheostosis: A Review of the Literature and a Case Report"

_medicina, 2023, doi:10.3390/medicina59050869_

Round 1
Reviewer 1 Report
The authors describe a new case of this uncommon disease, which lacks, unfortunately, specific diagnostic criteria.
General comments: Melorheostosis is a benign condition. Although it seems that when authors employ (repeatedly) the term "tumor" they refer to the meaning of this word as a non-specific growth, the term should be avoided (better, "lesion"), because it may be misleading. There are also some spelling/syntax errors (monostatic instead of monostotc, osteoids,...). In general, English should be improved.
Introduction:Regarding the structure of the manuscript, the 8 paragraphs of page 2 belong to the discussion. The objectives (I suppose last paragraph before the "case presentation") should be reorganized (the literature review is after the presentation of the case).
Case presentation: A table with some biochemical data, especially including CRP and/or other inflammatory markers/acute phase reactants (fibrinogen, albumin, ferritin, ..., if available), haemoglobin, alkaline phosphatse, etc, would be helpful. Did the patient show any alteration of the skin (soft tissue) over the bony lesion?. After the descriptoion of the immunohistochemical results, a sentence explaining the meaning of the results would be helpful (something like "therefore,... immunostaining was consistent with,...., and no data supported the existence of a malignant transformation..). The sentence relative to the possible complications of surgery (not presented by the patient) fits better into the discussion section.
Discussion: as previously suggested, it should not only include the comments about treatment, but also what the authors state in the introduction relative to the nature and clinical features of the disease, that should be transferred to the discussion. In addition, a differential diagnosis of the lesion should be performed , especially taking into account the non-specific radiological image of the presented case, somewhat atypical for Leri´s disease, and the importance to adopt therapeutic decisions destined to assure life maintenance, especially if the presence of/progression to a malignant tumor cannot be accurately ruled out.
Author Response
Thank you very much for the suggested modifications, all were adressed:
The authors describe a new case of this uncommon disease, which lacks, unfortunately, specific diagnostic criteria.
General comments: Melorheostosis is a benign condition. Although it seems that when authors employ (repeatedly) the term "tumor" they refer to the meaning of this word as a non-specific growth, the term should be avoided (better, "lesion"), because it may be misleading. There are also some spelling/syntax errors (monostatic instead of monostotc, osteoids,...). In general, English should be improved.
Thank you for the suggested modifications, there were all adressed.
Introduction:Regarding the structure of the manuscript, the 8 paragraphs of page 2 belong to the discussion. The objectives (I suppose last paragraph before the "case presentation") should be reorganized (the literature review is after the presentation of the case).
Thank you for the suggested modifications, there were all adressed.
Case presentation: A table with some biochemical data, especially including CRP and/or other inflammatory markers/acute phase reactants (fibrinogen, albumin, ferritin, ..., if available), haemoglobin, alkaline phosphatse, etc, would be helpful. Did the patient show any alteration of the skin (soft tissue) over the bony lesion?.
Thank you for the suggested modifications, but all the biochemical data were within normal limits, as we mentioned in the case presentation.
After the descriptoion of the immunohistochemical results, a sentence explaining the meaning of the results would be helpful (something like "therefore,... immunostaining was consistent with,...., and no data supported the existence of a malignant transformation..).
Thank you for the suggested modifications, there were all adressed.
The sentence relative to the possible complications of surgery (not presented by the patient) fits better into the discussion section.
Thank you for the suggested modifications, there were all adressed.
Discussion: as previously suggested, it should not only include the comments about treatment, but also what the authors state in the introduction relative to the nature and clinical features of the disease, that should be transferred to the discussion.
Thank you for the suggested modifications, there were all adressed.
In addition, a differential diagnosis of the lesion should be performed , especially taking into account the non-specific radiological image of the presented case, somewhat atypical for Leri´s disease, and the importance to adopt therapeutic decisions destined to assure life maintenance, especially if the presence of/progression to a malignant tumor cannot be accurately ruled out.
Thank you for the suggested modifications. The differential diagnosis with parosteal sarcoma, mixed metastases (considering the osteolytic and osteocondensating appearance), osteoma and myositis ossificans is required considering the non-specific radiological appearance. Taking into account all clinical, imaging and histopathological aspects, segmental resection with oncological limits and reconstruction with the help of a tumor prosthesis lead to a good survival and avoiding the simultaneous existence of a malignant bone lesion.
Reviewer 2 Report
The report introduced a good data about the disease, but the case came with unusual treatment option
As regards the abstract:
" An association with other benign or malignant bone lesions is also possible, and associations with osteosarcoma, malignant fibrous histiocytoma, or Buschke–Ollendorff syndrome have also been cited." You mean coincidence or overlapping symptoms with other differential diagnoses.
How your report affect the early recognition of these lesions?
Details about the case are deficient, cause of admission, manifestation, multiplicty of lesion, radiological evaluation, follow up and outcome , abstract should act independent to present the case
As regards the main text,
These lesions are usually incidental finding, so the problem is with those that were documented due to abnormal manifestations, better paraphrase this" given the limited number of cases worldwide"
Why your case was unique? This should be defined in the aim.you cited a report of 40 cases.
"was also cited in the specialist literature " needs paraphrasing, specialist?!,the word cited repeated many times better replaced by reported.
Images legend should provide a full description of the finding.
Repetition regards where the case was operated in the aim and case presentation
stiffening better stiffness
"antero-posterior and profile incidence was performed" , profile incidence needs paraphrasing
Dimensions should be defined in which plan?
"and marked functional impotence." Needs paraphrasing
Symptoms were very short term and these lesions are long standing one , lack of malignant transformation requires revision of decision regards the surgical intervention.
"with the increased potential to cause compression of the vessels and the femoral nerve, the lack of changes in the soft tissues" this is assumption not based on actual finding, bigger lesions than you have may not manifest.
The rational for surgical intervention is not strong, even the choice of surgery is very extreme.
You don't have complications, so these what was mentioned is the difficulties that may be encountered.
What was the follow up period?
Fig 10 at which follow up?
You mentioned many conservative lines , would it better to try bisphsphonate, denosumab before going to surgical intervention?
Author Response
The report introduced a good data about the disease, but the case came with unusual treatment option
As regards the abstract:
" An association with other benign or malignant bone lesions is also possible, and associations with osteosarcoma, malignant fibrous histiocytoma, or Buschke–Ollendorff syndrome have also been cited." You mean coincidence or overlapping symptoms with other differential diagnoses.
Thank you for suggested modifications, there were all adressed. The phrase mentioned above refers both to the coexistence of another benign or malignant bone pathology, but cases of malignant transformation into osteosarcoma or malignant fibrous histiocytoma of a pre-existing lesion of melorheostosis are also cited.
How your report affect the early recognition of these lesions?
Unfortunately, the non-specific radiological appearance could not establish the diagnosis of melorheostosis, but rather imposed the exclusion of a malignant proliferative process. Considering the periosteal reaction, specific to malignant bone tumors, highlighted on the radiological examination and computer tomography exam, the differential diagnosis with a benign lesion was secondary.
Details about the case are deficient, cause of admission, manifestation, multiplicty of lesion, radiological evaluation, follow up and outcome , abstract should act independent to present the case
Thank you for suggested modifications, there were all adressed.
As regards the main text,
These lesions are usually incidental finding, so the problem is with those that were documented due to abnormal manifestations, better paraphrase this" given the limited number of cases worldwide"
Thank you for suggested modifications, there were all adressed.
Why your case was unique? This should be defined in the aim you cited a report of 40 cases.
Thank you for suggested modifications, there were all adressed. Our case is unique due to the non-specific radiological appearance and the favorable functional result after the surgical treatment thus avoiding multiple therapeutic or surgical interventions with an adjuvant role and with an unpredictable functional result. As we can see in the clinical cases reported in the literature, the multiple surgical interventions with an adjuvant role can lead to unsatisfactory results in the long term and even to an evolution of the injury that no longer allows a surgical intervention with a curative role and to restore the functionality of the pelvic member thus so that in late cases, due to local complications, radical interventions such as amputation or disarticulation may be needed
"was also cited in the specialist literature " needs paraphrasing, specialist?!,the word cited repeated many times better replaced by reported.
Thank you for suggested modifications, there were all adressed.
Images legend should provide a full description of the finding.
Thank you for suggested modifications, there were all adressed.
Repetition regards where the case was operated in the aim and case presentation ???
stiffening better stiffness
Thank you for suggested modifications, there were all adressed.
"antero-posterior and profile incidence was performed" , profile incidence needs paraphrasing
Thank you for suggested modifications, there were all adressed.
Dimensions should be defined in which plan?
Thank you for suggested modifications, there were all adressed.
"and marked functional impotence." Needs paraphrasing
Thank you for suggested modifications, there were all adressed.
Symptoms were very short term and these lesions are long standing one , lack of malignant transformation requires revision of decision regards the surgical intervention.
Thank you for suggested modifications. The patient presented in the orthopedic department only when the pain was not relieved by the analgesic treatment in maximum doses. Being a chronic user of non-steroidal anti-inflammatory drugs, she cannot accurately identify the moment of worsening of the clinical symptoms and this is the reason why we could not specify in the text the period of onset of the symptoms.
"with the increased potential to cause compression of the vessels and the femoral nerve, the lack of changes in the soft tissues" this is assumption not based on actual finding, bigger lesions than you have may not manifest.
Taking into account the imaging appearance and the exophytic growth of the tumor that encases the femur 180 degrees, but also the nature of the compression pain, we consider that surgical treatment is an optimal option in the absence of conservative treatment results. I agree that larger melorostosis lesions can be asymptomatic, but probably in cases of lesions that do not exceed the cortical bone in such a manner.
The rational for surgical intervention is not strong, even the choice of surgery is very extreme.
You don't have complications, so these what was mentioned is the difficulties that may be encountered.
Thank for suggested modifications. We adressed all the complications as possible difficulties during and after surgery.
What was the follow up period?
Thank you for the question. The follow up period of the patient is 1 year.
Fig 10 at which follow up?
Thank you for suggested modifications, there were all adressed.
You mentioned many conservative lines , would it better to try bisphsphonate, denosumab before going to surgical intervention?
As we mentioned there are currently no guidelines for treatment based on scientific evidence, given the low number of cases diagnosed worldwide. The treatment must be adapted to each patient and requires a multidisciplinary team. Treatment decisions must be made based on the severity of the symptoms and the therapeutic possibilities. Considering the mixed and inconsistent results of the conservative treatment presented in other scientific articles, we considered that the failure of the first-line conservative treatment, the risk of evolution of the lesion, together with the patient, we decided that the lesion requires a surgical approach with curative and to improve the quality of life.
Given the limited number of cases worldwide and thus the lack of clinical guidelines regarding specialized treatment, melorheostosis remains an incompletely understood disease and what we can do until the formation of a significant batch of patients from a statistical point of view is to make a clinical judgment and to decide together with our patients the choice of the best treatment that will lead to an increased quality of life.
Round 2
Reviewer 1 Report
Most questions have been adequately adressed. It is not mandatory for acceptation, but given the rarity of the disease, it is advisable that the authors provide a table with the biochemical data available (especially thinking in potential future review papers about this uncommon disease).
Author Response
Comments and Suggestions for Authors
Most questions have been adequately adressed. It is not mandatory for acceptation, but given the rarity of the disease, it is advisable that the authors provide a table with the biochemical data available (especially thinking in potential future review papers about this uncommon disease).
Thank you for the suggested modifications, there were all adressed.
Reviewer 2 Report
The patient’s symptoms were mainly determined by the increase in volume of the tumor and the mass effect on the adjacent tissues, especially on the vessels and the femoral nerve. No oncological changes in the thoracic, abdominal, and pelvic regions”
How you detected the volume increament, oncological changes better replaced by other sites affection or metastasis , is there any type of ischemia regards the vessels affection?
“at the level of the thigh. Intervention” needs full stop adjustment
For how long the patient received conservative ttt, then you consider it failure?
Better to present your results after a fair follow up period ,45 days is very short period to determine the outcome.
Gender and 1 Year follow up mentioned in the response but not in the abstract or the case presentation.
Author Response
Comments and Suggestions for Authors
The patient’s symptoms were mainly determined by the increase in volume of the tumor and the mass effect on the adjacent tissues, especially on the vessels and the femoral nerve. No oncological changes in the thoracic, abdominal, and pelvic regions”
How you detected the volume increament, oncological changes better replaced by other sites affection or metastasis , is there any type of ischemia regards the vessels affection?
Thank you for the suggested modifications. The increase in volume of the lesion and the compressive effect on the adjacent soft tissues was objectified during the CT examination which, in addition to the chest, abdomen and pelvis, also included the thigh region. The mass effect on the vessels and the nerve was an indirect one through the compression of the muscle compartments. At the time of admission, the patient had no signs of ischemia in the lower limb.
“at the level of the thigh. Intervention” needs full stop adjustment
Thank you for the suggested modifications. There were all adressed.
For how long the patient received conservative ttt, then you consider it failure?
The patient was included in a full medical recovery program for eight weeks in a specialized medical center, during which she also received anti-algesic treatment in maximum doses, but without improvement regarding symptoms. Taking into account the chronic evolution of the pain, the complete lack of response to conservative treatment after 8 weeks and taking into account the lack of treatment guidelines in the case of melorheostosis, we consider that a surgical procedure is required to improve the patient's quality of life and restore the functionality of the affected limb.
Better to present your results after a fair follow up period ,45 days is very short period to determine the outcome.
Thank you for the suggested modifications. There were all adressed.
Gender and 1 Year follow up mentioned in the response but not in the abstract or the case presentation.
Thank you for the suggested modifications. There were all adressed.